# Hereditary Women’s Cancer: Management and Risk-Reducing Surgery

**DOI:** 10.3390/medicina59020300

**Published:** 2023-02-06

**Authors:** Carmine Conte, Silvia Pelligra, Giuseppe Sarpietro, Giuseppe Dario Montana, Luigi Della Corte, Giuseppe Bifulco, Canio Martinelli, Alfredo Ercoli, Marco Palumbo, Stefano Cianci

**Affiliations:** 1Department of General Surgery and Medical-Surgical Specialties, Institute of Obstetrics and Gynecology, A.O.U. Policlinico Rodolico, San Marco, University of Catania, 95125 Catania, Italy; 2Department of Woman and Child Health and Public Health, Catholic University of the Sacred Heart, 00168 Rome, Italy; 3Department of Neuroscience, Reproductive Sciences and Dentistry, School of Medicine, University of Naples Federico II, 80131 Naples, Italy; 4Department of Public Health, University of Naples Federico II, 80131 Naples, Italy; 5Unit of Gynecology and Obstetrics, Department of Human Pathology of Adult and Childhood “G. Barresi”, University of Messina, 98121 Messina, Italy

**Keywords:** risk-reducing surgery, abdominal hysterectomy, bilateral salpingo-oophorectomy, hereditary cancers, BRCA syndrome

## Abstract

Hereditary women’s syndromes due to inherited mutations result in an elevated risk of developing gynecological cancers over the lifetime of affected carriers. The BRCA 1 and 2 mutations, Lynch syndrome (LS), and mutations in rare hereditary syndromes increase this risk and require more effective management of these patients based on surveillance and prophylactic surgery. Patients need counseling regarding risk-reducing surgery (RRS) and the time required to perform it, considering the adverse effects of premenopausal surgery and the hormonal effect on quality of life, bone density, sexual activity, and cardiological and vascular diseases. Risk-reducing salpingo-oophorectomy (RRSO) is the gold standard for BRCA-mutated patients. An open question is that of endometrial cancer (EC) risk in patients with BRCA1/2 mutation to justify prophylactic hysterectomy during RRSO surgical procedures. RRS provides a 90–95% risk reduction for ovarian and breast cancer in women who are mutation carriers, but the role of prophylactic hysterectomy is underinvestigated in this setting of patients. In this review, we evaluate the management of the most common hereditary syndromes and the benefits of risk-reducing surgery, particularly exploring the role of prophylactic hysterectomy.

## 1. Introduction

Pathogenic variants in genes, usually associated with the regulation of cellular replication and double-strand DNA break repair, are often associated with hereditary cancers (HCs). 

BRCA mutations and Lynch syndrome are the most common hereditary conditions associated with cancers. BRCA1/2 syndrome involves an increased risk, higher than that of the general population, for female and male breast cancer, ovarian cancer (including fallopian tube and primary peritoneal cancers), and prostate and pancreatic cancer [1]. Lynch syndrome is an autosome-dominant condition characterized by DNA mismatch repair (MMR) gene mutations that predispose people of the same family to colorectal and endometrial cancer.

Genetic counseling HCs should be recommended in cases of individual suspected risk, based on an evaluation of the family history. Identifying women with an inherited risk of gynecological cancer gives the possibility to efficiently prevent cancer in patients and their blood relatives [2]. 

Screening and use of the oral contraceptive pill, breastfeeding, and a healthy lifestyle might decrease the risk of inherited cancers. However, for high-risk women, the main treatment opportunity is risk-reducing surgery (RRS) [1]. 

The term ‘risk-reducing’ is usually adopted rather than ‘prophylactic’ as it considers that not all malignant neoplasia can be prevented. Indeed, risk-reducing salpingo-oophorectomy (RRSO) cannot prevent primary peritoneal cancer in BRCA1/2-mutated women. Furthermore, RRSO could reduce, but not eradicate the breast cancer (BC) risk.

RRS requires complex decision making. The risk-reducing surgery types depend on a woman’s risk profile. The decision should be timed according to the woman’s age-dependent risk level, considering her pregnancy desire and willingness to take hormonal replacement therapy. RRSO will lead to fertility loss and surgical menopause in younger women. Furthermore, bone fracture risk, quality of life, and severe adverse events for RRSO or the effects of RRSO should be considered. However, delaying surgery increases the risk of gynecological malignancies.

## 2. Hereditary Women’s Cancer

### 2.1. BRCA Syndrome

The BRCA1 and BRCA2 tumor suppressor genes play a role in the homologous recombination repair (HRR) of double-stranded DNA breaks. Poly ADP ribose polymerase (PARP) is an essential component of single-strand DNA repair, and the inhibition of PARP increases double-strand breaks and prevents HRR-deficient (HRD) tumor cells from surviving chemotherapy-induced DNA damage, leading to synthetic lethality. In the general population, the carrier rate for BRCA1/2 mutations is estimated at 1:500–1:800. Some ethnic and geographical populations show a higher risk, for example, the Ashkenazi Jewish community, where BRCA1/2 mutations occur in up to 1 in 40 individuals [3]. 

A study investigating women diagnosed with ovarian cancer and its association with family history showed that the probability of carrying a gBRCA mutation in patients without family history is 14%; in patients with at least one relative with ovarian cancer, it is 45% and 47% if other family members have developed breast cancer. If breast and ovarian cancer are diagnosed in the family, the probability of carrying germline mutations of BRCA1 or 2 is 60% [4].

Women with a pathogenic BRCA1 mutation exhibit a 55–72% lifetime risk of developing BC that usually has an early onset and triple-negative subtype, with more aggressive and poorer prognosis. For BRCA2 mutations, the lifetime risk of breast cancer is 45–69%. 

BRCA-associated syndrome is characterized by an earlier onset than that of sporadic ovarian cancer, with a median age of 51 years [4]. The lifetime risk of developing ovarian cancer (OC) is 39–44% for BRCA1 and up to 11–17% for BRCA2-mutated women, with an onset 5–10 years later [5,6,7,8] (Table 1).

Because the fallopian tubes and endometrium share the paramesonephric (Müllerian) ducts as an embryological precursor, it is possible that the endometrium may be susceptible to analogous carcinogenesis. However, it is not clear if BRCA1 and BRCA2 mutations also provide increased lifetime risk of endometrial cancer (EC). 

In the literature, the data are conflicting. A large multicentric study documented a two–three-fold increased risk for EC in overall BRCA1 mutation carriers (SIR = 3.51, 95% CI = 2.61 to 4.72) and BRCA2 (SIR = 1.70, 95% CI = 1.01 to 2.87), with highest risk observed for the rare subgroups of serous-like and p53-abnormal EC in BRCA1 mutation carriers [9]. Furthermore, this study demonstrated that the risk depended on gBRCA1/2 mutations rather than previous hormonal therapy [9,10]. 

It is necessary to clarify this point because the serous/serous-like subtype accounts for only about 10% of EC patients but more than 40% of deaths in this subgroup of women [10]. However, previous studies did not report a raised EC risk or only an increased risk in an aggressive subgroup of EC with serous-like histology [12,13,19,20,21]. 

Although most studies documented a two-fold increase in risk relative to that of the general population [19,20,22,23], the results were statistically significant in only two of these studies [19,20].

In line with a previous study [9], a recent series advised a higher relative risk (RR) only in serous uterine cancer, but the results were only statistically significant for BRCA1-mutated populations [24].

It was supposed that the apparent increase in EC risk is not associated with the BRCA1 or 2 mutation, but with previous breast-cancer-related tamoxifen treatment. Indeed, different authors considered tamoxifen a relevant factor in endometrial cancer risk [9].

These uncertain data in the literature can be attributed to the small cohort sizes of the study, limited number of ECs, low median age at enrolment, relatively short follow-up, and lack of outcome validation [9].

In contrast with these data, a large retrospective study from the Breast Cancer Linkage Consortium documented an increased risk of EC in BRCA1-mutated women but not in BRCA2-mutated women [19]. 

A meta-analysis of 11 studies reporting a total of 13,871 carriers of BRCA1 and BRCA2 mutations suggested a slightly increased risk of EC, mainly for BRCA1. In particular, EC prevalence was 0.62% and 0.47% among BRCA1 and BRCA2 mutated population carriers, respectively, with an RR of 1.18 (95% CI 0.7–2.0) [25].

### 2.2. HRD Mutation

A particular mention should be made of the Homologous recombination status that may provide information on patients without a BRCA1/2 mutation. Homologous recombination repair (HRR) is a DNA repair pathway that acts on DNA double-strand breaks and interstrand cross-links. A deficiency in the HRR pathway (homologous recombination deficiency (HRD)) is a phenotype characterized by a cell’s inability to effectively repair DNA double-strand breaks using the homologous recombination repair (HRR) pathway. It has been associated with several tumor types including breast, ovarian, prostate, and pancreatic cancers. Tumors that are not HRD are termed homologous recombination proficient (HRP). Testing for a deficient HRR pathway is performed by probing the genome for evidence of genomic abnormalities. Several breast and ovarian cancer studies have identified genomic signatures of instability associated with an HRD phenotype [26].

Tumors with loss-of-function genes in this pathway might raise PARP inhibitors and platinum-based chemotherapy sensitivity. RAD51 testing showed high concordance with sBRCA mutation and genomic HRD. It should be incorporated into clinical decision making [26].

### 2.3. Lynch Syndrome

Lynch syndrome (LS) is a dominant inheritance due to pathogenic germline variants in the DNA mismatch repair (MMR) genes MLH1, MSH2, MSH6, and PMS2. LS causes a cancer predisposition defined by the early occurrence of colorectal cancer (CRC) and no CRCs, and EC is the most frequent. 

In a meta-analysis of 53 studies, including 12,633 EC patients, the authors estimated that the prevalence of LS in EC patients is approximately 3%, like that of CRC patients, and that about 10% of mismatch repair deficient (MMRd)/microsatellite unstable ECs are causally related to germline mutations of one of the MMR genes MLH1, PMS2, MSH2, and MSH6 [27].

ECs linked to MMR variants are usually diagnosed in younger individuals relative to the general population.

Carriers of different MMR variants exhibit distinct patterns of cancer risk and survival. LS confers an approximate 50–80% lifetime risk of CRC in either sex, and it confers an about 50% lifetime risk of EC in women. A higher risk of other cancers, including ovarian, small intestinal, brain, skin, and urinary tract, have also been described, without a significantly increased risk of BC in LS.

Identification of LS is recommended in all endometrial cancer patients, evaluating immunohistochemistry for MMR proteins, unless loss of MLH1 expression is caused by MLH1 promoter hypermethylation, which is likely due to somatic MLH1 mutations within the tumor itself [14].

Universal screening of EC patients for LS has been recommended by numerous experts and specialist societies. The results of a meta-analysis by Rian et al. may be beneficial in informing the planning and implementation of universal LS screening in EC patients [15,16,17,18,27,28].

The Manchester International Consensus Group strongly recommends that all women be informed of their MMR pathogenic-variant-specific risk of gynecological cancer, specifically endometrial and ovarian cancer, interpreted in the context of their family history (grade C) [18]. The Consensus Group recommends that women at risk of Lynch syndrome who have not experienced gynecological cancer undergo optional annual review from the age of 25 years with an appropriate clinician to discuss red-flag symptoms for endometrial and ovarian cancer where contraceptive and fertility needs are raised [18].

### 2.4. Other Genetic Syndromes

Cowden syndrome (CS), or PTEN hamartoma tumor syndrome, is an autosomal-dominant mutation in the PTEN gene. This gene is involved in cell signaling pathways for cell proliferation and survival. In this syndrome, there is the risk of neoplasia in different organs, including the endometrium, breast, thyroid, colon, and melanoma [29].

Endometrial cancer occurs in 21–28% of CS women and could grow in very young patients [30].

Germline variants of POLD1 and POLE, DNA replication, and proofreading polymerases have been identified as inherited predisposition in CRC and susceptibility to endometrial cancer. However, no study has quantified the EC risk to date [31].

MUTYH-associated polyposis is an autosomal-recessive predisposition to adenomatous polyposis and CRC. Patients with MAP are at higher risk of CRC (about 75%) and have a two-fold increased risk for EC [31]. However, the evidence is limited, and more extensive studies are needed. 

A mutation of the NTLH1 gene predisposes the individual to adenomatous polyposis and CRC. A study identified a germline variant of this gene in patients from three unrelated families, including women, where an EC occurred [32]. Another study evaluating 17 families with NTLH1 mutation described the occurrence of polyposis, BCs, and EC [33]. According to these results, NTHL1 deficiency determines a high-risk hereditary multitumor syndrome that seems to predispose to CRC, BC, and EC.

## 3. Management

### 3.1. Screening and Gynecologic Surveillance in Families with Hereditary Syndromes

Cancers in women with HS are mostly diagnosed at a younger age compared with patients in the general population. Actually, it is necessary to investigate new screening algorithms in order to mainly focus on the most common hereditary syndrome.

The most frequent tumors in BRCA pathogenic variant carriers are BC and OC in women and prostate cancer in men. International guidelines recommend regular screening for these cancers. In those patients, from the age of 25 years, BC screening with breast examination is recommended. Moreover, annual breast magnetic resonance is recommended for women 25–29 years of age, adding mammography in patients older than 30 years. Transvaginal ultrasound and CA125 blood testing are suggested for ovarian cancer screening from the age of 30 years [34].

The results of three studies suggest a potential stage shift when a risk of ovarian cancer algorithm (ROCA)-based ovarian cancer screening protocol is followed in high-risk women, though it remains unknown whether this screening protocol impacts survival [35,36,37]. In particular, the UK Collaborative Trial of Ovarian Cancer Screening (UKCTOCS), which assessed multimodality screening with transvaginal ultrasound (TVUS) and CA-125 versus either TVUS alone or no screening, showed that multimodality screening is more effective at detecting early stage cancer; however, after a median of 11 years of follow-up, a significant mortality reduction was not observed [35].

For those who have not elected RRSO, which is the recommended risk management option for ovarian cancer in carriers of a pathogenic, likely pathogenic (P/LP) BRCA1/2 variant, TVUS and serum CA-125 may be considered at the clinician’s discretion starting at 30 to 35 years of age [38].

In endometrial cancer, screening for genetic mutations should be considered, especially for patients < 50 years of age [38,39,40,41,42,43,44]. If these patients have Lynch syndrome, they are at a higher lifetime risk (≤60%) of endometrial cancer; thus, close monitoring and discussion of risk-reducing strategies are recommended [39,40,45,46,47,48]. In addition, their relatives may have Lynch syndrome. For patients and family members with Lynch syndrome but without endometrial cancer, a yearly endometrial biopsy is recommended to assess for cancer [43,49].

EC is considered a “sentinel” cancer in women with Lynch syndrome because it precedes other tumors, and it allows the recognition of other family members with mutations in MMR genes. Immunohistochemical (IHC) staining for MMR proteins (MLH1, MSH2, MSH6, and PMS2) in tumor specimens could be a screening test and is highly concordant with microsatellite instability testing in EC [18]. MMR protein loss or high microsatellite instability is diagnosed in about 20–35% of unselected ECs.

If immunohistochemistry discovers an MMR deficiency, next-generation sequencing (NGS) and germline testing are performed to evaluate the hereditary predisposition in LS patients. This strategy effectively identifies mutation carriers with the lowest number of diagnostic tests [14].

In patients with Cowden and other HSs with higher EC risk, the diagnosis is rare in an unscreened EC population due to their low prevalence. The search for genomic alteration seems helpful in the case of individual or family history [24]. 

### 3.2. Prophylactic Surgery 

Several clinical trials proved the cancer-preventing effects of the -reducing surgery [50,51]. 

Hysterectomy and bilateral salpingo-oophorectomy to prevent ECs and OCs should be preferably offered after childbearing and earlier than 40 years of age. The surgeon may discuss with the patient all the pros and cons of prophylactic surgery, including the risk of occult gynecological cancer diagnosis after the prophylactic surgical procedures. Indeed, women do not present symptoms before the prophylactic surgery, but with occult neoplasia, might be detected in this high-risk population. 

Surgery is not without risk and potential long-term side effects, however, and preoperative counseling is important. The laparoscopic approach is associated with less postoperative pain, quicker recovery, and improved short-term quality of life, making it the preferred approach in uncomplicated cases. Surgical menopause follows risk-reducing oophorectomy in premenopausal women. This is associated with vasomotor symptoms, urogenital dryness and atrophy, reduced sexual function, emotional lability, and cognitive decline, as well as increased risks of osteoporosis, cardiovascular disease, and CRC. Thus, prescription of estrogen-only hormone replacement therapy (HRT) until at least natural menopause age (~51 years) is strongly recommended to prevent these sequelae [52,53,54].

Previous series showed more favorable oncological outcomes in terms of overall survival among carriers of a pathogenic BRCA1/2 variant in women with OC than in noncarrier patients. Moreover, survival outcomes seem to be most favorable for carriers of a pathogenic BRCA2 variant. Additionally, BRCA2 mutations were associated with significantly better response rates (compared with noncarriers or BRCA1 mutation carriers) to first-line chemotherapy. Conversely, the BRCA1-mutated variant did not result in better prognosis or higher chemotherapy response [55]. 

In case of women carrying a pathogenic BRCA1/2 variant who underwent RRSO, the authors reported a range of 4–9% of occult gynecological malignancy. Tubal intraepithelial carcinoma (TIC) is considered an early precursor lesion for serous OCs. In the case of patients with a pathogenic BRCA1 or BRCA2 variant, the risk of TIC (with or without other lesions) was about 5–8% of cases from women who underwent RRSO [56]. 

In the case of early cancers in women with a pathogenic BRCA1/2 variant, the predominant site of the origin of the disease is the fimbriae or the distal part of the tube. In order to detect potential occult lesions, both ovaries and fallopian tubes should be removed and carefully sectioned with microscopic examination. Moreover, international guidelines suggest adding surgical procedures such as a laparoscopic examination of all peritoneal sites and pelvic washing to identify other sites of occult metastasis [57,58]. 

The occult disease diagnosis is more frequent in BRCA1-mutated patients than BRCA2 carriers (4.2% vs. 0.6%, respectively). Women with BRCA1 mutations are more likely to develop malignant pathology at a younger age, so one would expect an increase in occult cancers at the time of RRSO compared with BRCA2 carriers [56]. 

Although RRSO is the gold standard of prophylactic management for BRCA-mutated women, the benefit of concomitant hysterectomy (H) remains controversial [23].

An aspect to consider is that although hysterectomy is not thought to be justified for cancer prevention in women with BRCA1 or 2 mutations, it can simplify later hormonal therapy to decrease the risk of BC or estrogen for menopausal symptoms [59]. Indeed, in case of RRSO plus hysterectomy, the patient is candidate for estrogen-alone hormone replacement therapy (HRT), instead of combined therapy with estrogen and progesterone, which is indicated when the uterus is not removed. Estrogen-alone therapy is associated with lower risk of breast cancer than combined therapy. 

For patients who choose to undergo prophylactic surgery, the surgeon may inform the woman about the risks and benefits of concurrent total hysterectomy [38]. 

In BRCA1 mutation carriers, delaying RRSO beyond 40 years is associated with an increased risk of developing OC or tubal cancer. The age of onset in BRCA2 mutation carriers is later and the penetrance is lower than in BRCA1 carriers, so surgery may be delayed until after 40 years. If a woman’s mutation status is not known before prophylactic surgery, genetic counseling and testing should be suggested to the patient. If this is not possible, then the time decision needs to be made taking into account the family history. 

Those with Cowden and Lynch syndrome might postpone surgical procedures until after forty years due to the median age of the onset of EC and the lower penetrance for ovarian cancer in Lynch syndrome. 

A single-institution study of Duenas analyzed 976 LS individuals with a mean follow-up of 10.2 years; of 531 women with LS, risk-reducing surgery significantly reduced EC (25.2 vs. 9.1%), demonstrating that risk-reducing surgeries are effective in decreasing the incidences of colorectal and gynecological cancer in LS carriers [60]. 

Identification of a pathogenic variant (PV) in an ovarian cancer-risk gene may initiate more intensive and personalized medical management that would not be prompted based on family history alone.

The age of OC incidence in patients carrying BRIP1 and RAD51C/D mutations is under investigation [60,61,62,63,64].

The absolute ovarian cancer risk is lower than in BRCA carriers, with lifetime risks ranging from 6 to 15%.

For women with PVs in BRIP1, RAD51C, or RAD51D, NCCN guidelines recommend that RRSO be considered at age 45–50 [38] (Table 2).

In a recent study, Cummings et al. reported a median age at ovarian cancer diagnosis of 53 years for BRCA1, 59 years for BRCA2, 65 years for BRIP1, 62 years for RAD51C, and 57 years for RAD51D [60]. This study suggested that it is safe to delay RRSO until age 45–50 in RAD51D PV carriers and possibly until age 50–55 in BRIP and RAD51C PV carriers [60].

## 4. Conclusions

Risk-reducing surgery should be offered to women with a high-risk genetic mutation or a strong family history. Accordingly, a general recommendation regarding risk-reducing surgery, particularly for premenopausal patients, cannot be given. Surgical procedures should be individually tailored based on the type of cancer the patient is at risk of and the patient’s age, future need for hormonal replacement treatment, history of breast cancer, tamoxifen use, and personal operative risks.

Nowadays, risk-reducing surgery can be successfully achieved by minimally invasive surgery. If it is performed by a qualified surgeon, it has a minimal risk of postoperative complications, and short operative time, hospital stay, and recovery [65,66,67]. However, controversy about performing a hysterectomy at the time of RRSO is still ongoing [68]. 

Considering recent studies that demonstrated the increased risk of EC in BRCA1/2 carriers and the development of surgical skill in minimally invasive surgery, we suggest evaluating the removal of the uterus at the time of the RRSO to avoid a second operation in case of occult endometrial malignancy.

## Figures and Tables

**Table 1 medicina-59-00300-t001:** Risk of malignancy in individuals with/without a germline pathogenic variant.

Cancer Type	General Population Risk	Risk of Malignancy ^1^				
		BRCA1	BRCA2	MSH2	MLH1	MSH6
Lifetime risk of OC	1–2%	39–44%	11–17%	10–17%	10–15%	10–13%
Cumulative risk of OC at 40 ys of age	<1%	1.5%	<1%	4%	3%	4%
Lifetime risk of EC	<1%	2%	<2%	21–57%	34–54%	16–49%
Cumulative risk of EC at 40 ys of age	<1%	<1%	<1%	2%	3%	0%
Breast cancer	12%	55–72%	45–69%	n.a.	n.a.	n.a.

Abbreviations: OC = ovarian cancer, EC = endometrial cancer; n.a.= not available; ys = years. ^1^ References: [6,7,8,9,10,11,12,13,14,15,16,17,18].

**Table 2 medicina-59-00300-t002:** Summary of OC risk for RRSO by gene.

Gene	NCCN Recommendation [38]	Age at RRSO (Years)	RRSO vs. No RRSO According to BRCA Mutation Status [52]
BRCA1	RRSO	35–40	Overall survival: HR 0.30 (95% CI: 0.17 to 0.52)
BRCA2	RRSO	40–45	Overall survival: HR 0.44 (95% CI: 0.23 to 0.85)
MLH 1	Consider TH/RRSO	After childbearing, not earlier than 35–40	n.a
MSH 2	Consider TH/RRSO	After childbearing, not earlier than 35–40	n.a
MSH 6	Consider TH/controversial for RRSO but potentially beneficial	After childbearing, not earlier than 35–40	n.a
BRIP 1	Consider RRSO	45–50	n.a
RAD 51C	Consider RRSO	45–50	n.a
RAD 51D	Consider RRSO	45–50	n.a

Abbreviations: TH = total hysterectomy; RRSO = risk-reducing bilateral salpingo-oophorectomy; CI = confidence interval; HR = hazard ratio; n.a. = not available.

## Data Availability

Not applicable.

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
