# Peer review of "Hereditary Women’s Cancer: Management and Risk-Reducing Surgery"

_medicina, 2023, doi:10.3390/medicina59020300_

Round 1
Reviewer 1 Report
It was a pleasure to read this manuscript on »Hereditary women's cancer: management and risk-reducing surgery«. There are a few inaccuracies and needs for clarification prior to consideration for publication of this manuscript.
Major considerations:
- The introduction does not specify what are hereditary women cancers – Consider adding a description what are the most common cancers linked to genetic alterations and their overall impact (what proportion of those cancers is attributed to hereditary conditions). Make sure, to specify which of the genes are connected with hereditary cancer eg. ovarian cancer (genes ...), endometrial cancer (genes ...)
- BRCA Syndrome chapter: The listed risk levels of BRCA are underestimated, recent literature puts the incidence at higher levels. Please, re-review the literature and after reporting on the incidence accurately reffer to the specific literature. Currently it seems very confusing in this chapter, as the most important cancers – ovarian/breast for BRCA have not been given a lot of space, yet endometrial cancer half a page. Consider restructuring it and also addressing: what type of genetic testing is needed – somatic/germline/both and why. The HRD section seems misplaced and will need more explanation as a separate chapter.
- Lynch Syndrome chapter: As this is a manuscript on womens cancers – more focus should be given to endometrial cancer: what is the incidence of Lynch syndrome in EC population, how much in MMRd population, how does this impact prevention?
- Other genetic syndromes chapter: This chapter is rather short and superficial. If you would like to give just short intros into these topics – consider restructuring the manuscript based on the caused cancers (eg EC, OC, BC)
- Managament:
o Please cite the guidelines actually recommending these prevention methods, as it does not seem up to date with our current knowledge: »The most frequent tumors in BRCA pathogenic variant carriers are BC and OC in women and prostate cancer in men. International guidelines recommend regular screen-ing for these cancers. In those patients, from the age of 25 years it is recommended BC screening with breast examination. Moreover, annual breast magnetic resonance is rec-ommended for women 25–29 years of age, adding mammography in patients older than 30 years. Transvaginal ultrasound and CA125 blood testing are suggested for ovarian can-cer screening from the age of 30 years«. Within a clinical setting TVUS+CA125 is not accepted as a screening method for OC.
o This section should be integrated with the previous mentions of genetic syndromes: »EC is considered a "sentinel" cancer in women with Lynch Syndrome because it pre-cedes other tumors, and it allows the recognition of other family members with mutations in MMR genes. In patients with Cowden and other HSs with higher EC risk, the diagnosis is rare in an unscreened EC population due to their low prevalence. The search for genomic altera-tion seems helpful in the case of individual or family history [17]. Immunohistochemical (IHC) staining for MMR proteins (MLH1, MSH2, MSH6, and PMS2) in tumor specimens could be a screening test and is highly concordant with mi-crosatellite instability testing in EC [27]. MMR protein loss or high microsatellite instabil-ity is diagnosed in about 20–35% of unselected ECs. If immunohistochemistry discovers an MMR-deficiency, next-generation sequencing (NGS) and germline testing will be performed to evaluate the hereditary predisposition in LS patients.«
o Risk reduction section:
§ »The age of OC incidence in patients carrying BRIP1 and RAD51C/D mutations is cur-rently uncertain.« - I would state this with very much caution, as BRIP1/RAD51C and RAD51D are accepted with a treshold of more than cca. 10 % risk of lifetime OC leading to acceptability for RRSO. Please refer to the literature:
· https://ovarianresearch.biomedcentral.com/articles/10.1186/s13048-021-00809-w
· https://www.frontiersin.org/articles/10.3389/fonc.2022.1030786/full
· https://www.mdpi.com/2072-6694/13/17/4344
§ Please ensure you list the main studies in a table currently ongoing on RRSO, also consider the impact of RRESDO.
Minor considerations:
- Abstract: The statement of the impact of hysterectomy not being assessed is inaccurate. There have been assessment of risk reduction and even cost-effectiveness. Please review the literature again.
· Additionally, not a lot of focus has been given to breast cancer. Either consider excluding it or focusing more on it. Both approaches can be valid. Please also re-evalaute the following lit in terms of breast cancer sussceptability genes: 10.1056/NEJMoa1913948
I believe this manuscript could have, with substantial changes a contribution to the current body of knowledge.
Author Response
It was a pleasure to read this manuscript on »Hereditary women's cancer: management and risk-reducing surgery«. There are a few inaccuracies and needs for clarification prior to consideration for publication of this manuscript.
Major considerations:
- The introduction does not specify what are hereditary women cancers – Consider adding a description what are the most common cancers linked to genetic alterations and their overall impact (what proportion of those cancers is attributed to hereditary conditions). Make sure, to specify which of the genes are connected with hereditary cancer eg. ovarian cancer (genes ...), endometrial cancer (genes ...)
Thank you for the revision
As suggested by the Reviewer we better specified the most common hereditary cancers and correlated genes in the introduction paragraph. We did not highly enhance the BRCA and LS in the introduction because we discussed it later in the related paragraph.
Please, see the revised manuscript.
- BRCA Syndrome chapter: The listed risk levels of BRCA are underestimated, recent literature puts the incidence at higher levels. Please, re-review the literature and after reporting on the incidence accurately reffer to the specific literature. Currently it seems very confusing in this chapter, as the most important cancers – ovarian/breast for BRCA have not been given a lot of space, yet endometrial cancer half a page. Consider restructuring it and also addressing: what type of genetic testing is needed – somatic/germline/both and why. The HRD section seems misplaced and will need more explanation as a separate chapter.
As correctly pointed out by the Reviewer, we better discussed the ovarian cancer correlated to BRCA syndrome.
As mentioned in the introduction paragraph we included the cancer risk incidence correlated to BRCA syndrome.
We did not review the breast cancer given topic of this special issue about the abdominal hysterectomy.
Moreover, we better highlighted the HRD syndrome.
- Lynch Syndrome chapter: As this is a manuscript on womens cancers – more focus should be given to endometrial cancer: what is the incidence of Lynch syndrome in EC population, how much in MMRd population, how does this impact prevention?
As suggested by the Reviewer we implemented the LS paragraph with the incidence of LS and screening into the population
Please, see revised manuscript.
- Other genetic syndromes chapter: This chapter is rather short and superficial. If you would like to give just short intros into these topics – consider restructuring the manuscript based on the caused cancers (eg EC, OC, BC)
- Managament:
o Please cite the guidelines actually recommending these prevention methods, as it does not seem up to date with our current knowledge: »The most frequent tumors in BRCA pathogenic variant carriers are BC and OC in women and prostate cancer in men. International guidelines recommend regular screen-ing for these cancers. In those patients, from the age of 25 years it is recommended BC screening with breast examination. Moreover, annual breast magnetic resonance is rec-ommended for women 25–29 years of age, adding mammography in patients older than 30 years. Transvaginal ultrasound and CA125 blood testing are suggested for ovarian can-cer screening from the age of 30 years«. Within a clinical setting TVUS+CA125 is not accepted as a screening method for OC.
As suggested by the Reviewer we re-reviewed screening managenent and we cited the international NCCN guidelines.
Studies assessing whether ovarian cancer screening procedures are sufficiently sensitive or specific have yielded mixed results.
See revised manuscript.
o This section should be integrated with the previous mentions of genetic syndromes: »EC is considered a "sentinel" cancer in women with Lynch Syndrome because it pre-cedes other tumors, and it allows the recognition of other family members with mutations in MMR genes. In patients with Cowden and other HSs with higher EC risk, the diagnosis is rare in an unscreened EC population due to their low prevalence. The search for genomic altera-tion seems helpful in the case of individual or family history [17]. Immunohistochemical (IHC) staining for MMR proteins (MLH1, MSH2, MSH6, and PMS2) in tumor specimens could be a screening test and is highly concordant with mi-crosatellite instability testing in EC [27]. MMR protein loss or high microsatellite instabil-ity is diagnosed in about 20–35% of unselected ECs. If immunohistochemistry discovers an MMR-deficiency, next-generation sequencing (NGS) and germline testing will be performed to evaluate the hereditary predisposition in LS patients.«
As suggested by the reviewer we integrated this paragraph.
Please, See revised manuscript
o Risk reduction section:
- »The age of OC incidence in patients carrying BRIP1 and RAD51C/D mutations is cur-rently uncertain.« - I would state this with very much caution, as BRIP1/RAD51C and RAD51D are accepted with a treshold of more than cca. 10 % risk of lifetime OC leading to acceptability for RRSO. Please refer to the literature:
- https://ovarianresearch.biomedcentral.com/articles/10.1186/s13048-021-00809-w
- https://www.frontiersin.org/articles/10.3389/fonc.2022.1030786/full
- https://www.mdpi.com/2072-6694/13/17/4344
As suggested by the Reviewer we re-revoewed the data about BRIP1 and RAD51C/D and added the suggested references.
Please, see revised manuscript.
- Please ensure you list the main studies in a table currently ongoing on RRSO, also consider the impact of RRESDO.
As suggested by the Reviewer we summerized the principal findings of RRSO in a table. Given the narrative type of this review,and not a sistematic review, we reported a summary of the ovarian cancer risk for RRSO by gene.
Please see table 2
Minor considerations:
- Abstract: The statement of the impact of hysterectomy not being assessed is inaccurate. There have been assessment of risk reduction and even cost-effectiveness. Please review the literature again.
As correctly pointed out by the Reviewer, we modified the sentence.
Please see the revised abstract.
- Additionally, not a lot of focus has been given to breast cancer. Either consider excluding it or focusing more on it. Both approaches can be valid. Please also re-evalaute the following lit in terms of breast cancer sussceptability genes: 10.1056/NEJMoa1913948
Thank you for the comment. We did not focus our review on breast cancer hereditary management. Given the special issue of abdominal hysterectomy, we highlighted women’s cancers requiring prophylactic hysterectomy.
I believe this manuscript could have, with substantial changes a contribution to the current body of knowledge.
Reviewer 2 Report
The review addresses hereditary female cancers (HFCs) and the benefits of risk-reducing surgery, particularly the role of prophylactic hysterectomy. The paper seems interesting to readers but needs more focus and clarification, namely the following:
1) A tabulation disclosing lifetime cancer risks (by organ) and germline pathogenic genes/variants in clinically validated HFCs;
2) Comments on the percentage of cases without a suggestive family history;
3) Tabulate validated interventions for prevention or early detection of future cancers;
4) Also, comment briefly on quality of life issues and complications following risk-reducing surgery.
Author Response
The review addresses hereditary female cancers (HFCs) and the benefits of risk-reducing surgery, particularly the role of prophylactic hysterectomy. The paper seems interesting to readers but needs more focus and clarification, namely the following:
1) A tabulation disclosing lifetime cancer risks (by organ) and germline pathogenic genes/variants in clinically validated HFCs;
As suggested by the Reviewer we summarized the principal findings of lifetime cancer risks.
Please see table 1
2) Comments on the percentage of cases without a suggestive family history;
As suggested by the Reviewer, we added this comment to the BRCA paragraph.
Please, see the revised manuscript (pages 5-6, lines 100-105).
3) Tabulate validated interventions for prevention or early detection of future cancers;
As suggested by the Reviewer we summarized the principal findings of validated interventions for prevention
Please see table 2
4) Also, comment briefly on quality of life issues and complications following risk-reducing surgery.
Thank you for the suggestion. As suggested by the Reviewer, we added a comment on the patients' quality of life and complications after RRS.
Please, see revised manuscript (page 13, lines 279-287)
Round 2
Reviewer 2 Report
I believe the paper in its current form may interest readers.